# Misfire Detection in Automotive Engines Using a Smartphone through Wavelet and Chaos Analysis

**DOI:** 10.3390/s22145077

**Published:** 2022-07-06

**Authors:** Nayara Formiga Rodrigues, Alisson V. Brito, Jorge Gabriel Gomes Souza Ramos, Koje Daniel Vasconcelos Mishina, Francisco Antonio Belo, Abel Cavalcante Lima Filho

**Affiliations:** 1Graduate Program in Mechanical Engineering (PPGEM), Federal University of Paraiba (UFPB), João Pessoa 58051-900, Brazil; nayara.rodrigues@ct.ufpb.br (N.F.R.); kojemishina@gmail.com (K.D.V.M.); belo@cear.ufpb.br (F.A.B.); abelima@gmail.com (A.C.L.F.); 2Graduate Program in Informatics (PPGI), Federal University of Paraiba (UFPB), João Pessoa 58051-900, Brazil; 3Department of Physics, Federal University of Paraiba (UFPB), João Pessoa 58051-900, Brazil; jorgephysics@gmail.com

**Keywords:** combustion engine, failure diagnosis, wavelet, SAC-DM

## Abstract

Besides the failures that cause accidents, there are the ones responsible for preventing the car’s motion capacity. These failures cause inconveniences to the passengers and expose them to the dangers of the road. Although modern vehicles are equipped with a failure detection system, it does not provide an online approach to the drivers. Third-party devices and skilled labor are necessary to manage the data for failure characterization. One of the most common failures in engines is called misfire, and it happens when the spark is weak or inexistent, compromising the whole set. In this work, two algorithms are compared, based on Wavelet Multiresolution Analysis (WMA) and another using an approach performing signal analysis based on Chaos using the density of maxima (SAC-DM) to identify misfare in a combustion engine of a working automotive vehicle. Experimental tests were carried out in a car to validate the techniques for the engine without failure, with failure in one piston, and with two failed pistons. The results made it possible to obtain the failure diagnosis for 100% of the cases for both WMA and SAC-DM methods, but a shorter time window when using the last one.

## 1. Introduction

When not previously detected, failures in motor vehicles can cause them to lose their locomotion capacity, causing financial losses, inconvenience to their passengers, and subjecting them to the inherent dangers of the roads. Currently, the automaker performs the diagnosis, which is responsible for an intense battery of tests on the vehicle, and by the dealerships, where the customer reports after the detection of a failure and has technical assistance. At the automaker, the engineers responsible have access to the variables of the systems available in the vehicle in real-time through software connected to the car via the OBD (On-Board Diagnostic) physical input. Detecting a defect is predominantly the owner’s responsibility (driver), except for inspections scheduled at the production. It constitutes a gap for studies to develop easy access and use products that do not demand a high degree of technical knowledge for ordinary users. The misfire event in an automotive engine is a common failure characterized by an abnormal condition of an engine due to partial or non-existent combustion of the fuel/air mixture during the development of mechanical power. This event is an unwanted phenomenon that compromises the normal functioning of the engine, increasing the pollution of the emitting gases and reducing its efficiency [1].

This work proposes a practical model for misfire diagnosis in internal combustion engines through the vibration signal of a smartphone accelerometer comparing two data processing techniques: Wavelet Multiresolution Analysis (WMA) and the SAC-DM (Signal Analysis based on Chaos using Density of Maxima [2,3].

For three operating conditions, experimental tests were carried out: the engine running perfectly, with a misfire in the first cylinder and a misfire in the first and fourth cylinders. In the following section, the most recent state-of-the-art works will be presented and compared with the proposed article to highlight the work’s contribution.

## 2. State of the Art

The works related to the diagnosis of misfire failures in automotive vehicles differ according to sensing and the data processing technique. In Table 1, the main works related to the topic are listed. In recent years, a trend towards diagnosis through vibration has been noticed [1,4,5,6,7], as opposed to classical methods based on the angular velocity of the engine [8,9,10]. Sensory methods based on the measurement of the exhaust gas temperature [11] and acquisition of the sound emitted by the engine [1,12] were also found.

Processing techniques involve approaches in the frequency and time domain involving Fast Fourier Transform (FFT), Wavelets, finite element techniques, artificial neural networks (ANN), different statistical models, or a combination of one or more methods, as can be seen in Table 1.

Wavelet is a powerful mathematical tool that has been used for decades in the field of signal processing, allowing hybrid time/frequency domain analyses, and has already been tried in the diagnosis of combustion engine failures, obtaining satisfactory results [12,13]. The SAC-DM technique was applied for the first time to diagnose faults in mechanical systems recently with good results [2,3].

Signal processing using wavelet is similar to the SAC-DM technique because it is a time-domain approach whose analysis variable is measured in frequency units (Hz). Therefore, in addition to the comparative analysis illustrated in Table 1, it was decided to use the Wavelet Multiresolution Analysis (WMA) signal analysis technique independently of the SAC-DM to compare the two techniques when used for fault diagnosis in combustion motors.

**Table 1 sensors-22-05077-t001:** Comparison between the main techniques for detecting misfire in internal combustion engines.

Work	Sensor	Technique	Accuracy
Hmida et al. (2021) [4]	Vibration	Torsional model of a four cylinder crankshaft	Not checked
Lima et al. (2021) [12]	Sound	Wavelet/fractal dimensions and ANN	99.58%
Firmino et al. (2021) [1]	Vibration/sound	FFT/ANN	99.30% to vibration and 98.70% to Sound
Du et al. (2021) [14]	Vibration	Sparse decomposition and engine finite element model	Not checked
Gu et al. (2021) [6]	Vibration	Multivariate Empirical Decomposition Mode, and Dispersion Entropy	Not Checked
Qin et. al. (2021) [7]	Vibration	Deep twin CNN with multi-domain inputs	97.019%
Du et al. (2020) [5]	Vibration	probabilistic neural network	100%
Zheng et al. (2019) [8]	Angular Velocity	ANN	Not verified
Jafarian et al. (2018) [15]	Vibration	FFT, ANN, SVM, and KNN	98%
Chen et al. (2015) [16]	Vibration	ANN	Not verified
Tamura et al. (2011) [11]	measurement of exhaust gas temperature	Statistical analysis of acquired curve behavior	Not verified
Rizvi et al. (2011) [9]	angular velocity	Markov Chain	Not verified
Hu et al. (2011) [10]	angular velocity	multivariate statistical analysis algorithm	More than 90%
Proposed Article	Vibration (smartphone)	Density of Maxima, wavelet, FFT	100%

In this way, we can highlight the following points of innovation and contribution of this work: (1) Application of the SAC-DM technique for the first time to identify faults in combustion engines presenting effectiveness compatible with the other techniques present in the literature, high sensitivity to initial conditions (characteristic of the presence of Chaos), and low computational effort; (2) Development of a WMA-based algorithm for misfire identification in combustion engines; (3) Compare the two processing techniques (SAC-DM and WMA).

## 3. Fundamentals

### 3.1. Multiresolution Wavelet Analysis

In multiresolution wavelet analysis (MWA), the signal is decomposed over time into an approximation, and several signals with different frequency compositions are called details, making it possible to obtain analyses in time and frequency. In this work, the WMA used is based on the Mallat algorithm [17]. In wavelet decomposition, the original sampled signal x[n] is submitted to a low-pass filter, which provides approximation A1, and by a high-pass filter, giving rise to detail D1, according to Equations (Equation 1) and (Equation 2).
(1)$$A\left(n\right)=\sum_{k}h\left(n-2k\right)x\left(n\right)$$
(2)$$D\left(n\right)=\sum_{k}g\left(n-2k\right)x\left(n\right)$$
where *k* is the level of decomposition and *h* and *g* are filters from the wavelet family (the Daubechies family was used in this work), low-pass and high-pass, respectively. From level *k* = 2, the high-pass and low-pass filters are applied to the approximation signals of the previous level, according to the Equations (Equation 3) and (Equation 4), forming a structure of cascading decomposition.
(3)$$A_{m}\left(n\right)=\sum_{k}h\left(n-2k\right)A_{m-1}\left(k\right)$$
(4)$$D_{m}\left(n\right)=\sum_{k}g\left(n-2k\right)D_{m-1}\left(k\right)$$

The structure of the decomposition of the signals performed by the filtering process defines the frequency range. We are applying these filters results in the signal being subsampled in half (downsampling), causing the frequency bandwidth to be halved at each filter output. Due to the decomposition process, the approximations and details have frequency components with a passband that obeys the relation of Equations (Equation 5) and (Equation 6), where $f_{s}$ is the sampling frequency of the signal.
(5)$$A_{k}=\left[0,\frac{f_{s}}{2^{k+1}}\right]$$
(6)$$D_{k}=\left[\frac{f_{s}}{2^{k+1}},\frac{f_{s}}{2^{k}}\right]$$

### 3.2. Density of Maxima

According to [18], the density of maxima can be obtained in terms of the autocorrelation function. Equation (Equation 7) is applied to the analysis of samples of non-periodic signals, where $\langle \rho_{i}\rangle$ is theoretical and calculated through the second and fourth derivatives of the function of autocorrelation at zero, as presented in [3].
(7)$$\langle \rho_{i}\rangle =\frac{1}{2\pi }\sqrt{\frac{q_{i}^{\prime \prime 2}}{q_{i}^{\prime 2}}}=\frac{1}{2\pi }\sqrt{\frac{\frac{d^{4}C_{i}}{dt^{4}}\left(0\right)}{\frac{-d^{2}C_{i}}{dt^{2}}\left(0\right)}}$$
where $C_{i}\left(t\right)$ is the autocorrelation function of the sign given by Equation (Equation 8).
(8)$$C_{i}\left(\delta t\right)=\langle q_{i}\left(t+\delta t\right),q_{i}\left(t\right)\rangle$$

In a short time evolution, the autocorrelation functions for chaotic systems can be approximated to a cosine function, which, when applied to the Equation (Equation 7) can be expressed according to the correlation length (τ), which is obtained according to the value of the half-height length, that is when the autocorrelation function first assumes the value equal to 0.5. As presented in [18] the value of the maximum density can be approximated in chaotic systems according to the Equation (Equation 9).
(9)$$\langle \rho \rangle =\frac{1}{6\tau }$$

For practical purposes, the chaotic property of the vibration signal used can be verified by comparing the value 〈ρ〉 of Equation (Equation 9) with the value obtained from the peak count of the signal that generates the correlation function, technique called SAC-DM (Signal Analysis based on Chaos using Density of Maxima) given by Equation (Equation 10).
(10)$$SACDM=\frac{numberofpeaks}{timewindow}$$

When proving the chaotic behavior of the signal, it is possible to observe the density of maximums as a deterministic tool, sensitive to the initial conditions of a system, whose specific application of this work can be correlated with the behavior of the motor running normally or with failure, through the counting of peaks in a specific time range of the vibration signal.

We have previously detected that SAC-DM has no unique range of values for every situation. As a fundamental characteristic of all Chaotic systems, SAC-DM is extremely sensitive to even tiny changes in its variables. Thus, for each initial configuration (engine specification, rotation, and fuel), one needs to collect the SAC-DM values to establish the common values. From this range, any disturbance in the system can be detected from changes in SAC-DM values.

## 4. Methods

The present work used the accelerometer sensor to measure the vibration signal. Currently, smartphones are manufactured with this integrated sensor, responsible for the screen rotation movement. Through the SPARKvue application installed on the cell phone, acceleration signals were acquired using a sampling frequency of 200 samples per second in tests lasting about 1 min.

Figure 1 illustrates the outline of the data analysis methodology in this work. The same acceleration signal was subjected to a processing technique using wavelet multiresolution analysis (WMA) and the SAC-DM technique.

The layout of the WMA and the decomposition quantity of the details depend on the type of application, the frequency range of interest, and the acquisition rate used. The vibration signal was decomposed to detail D3 using the WMA technique. The choice of this signal is since the frequency components in which the faults are related to the acyclism frequency of the car (26.30 Hz) and the engine rotation frequency (13.15) running in neutral. From Equation (Equation 6), one can see that both frequencies are close to the passband of detail D3 (25–12.5 Hz), considering the sampling rate of 200 samples per second. From the calculation of the standard deviation of the signal, it is possible to identify the state of the motor.

Before applying the SAC-DM technique, it is necessary to prove the chaotic behavior of the engine, which has already been done in [12] for the sound signal. The diagnosis is performed by verifying the change in the pattern of the number of peaks of the vibration signal over time, using Equation (Equation 10).

## 5. Results

A smartphone was used as a failure detection instrument to acquire acceleration samples, storage, and data presentation. The Android Smartphone used in the experiment was the Samsung Galaxy J7 Prime model. The reading range of a typical smartphone accelerometer is ±2*g*, where *g* is the gravity acceleration, measured based on an inertial frame of reference. The Figure 2 shows the smartphone positioned on top of the internal combustion engine minimally fixed with adhesive tapes.

A Ford Fiesta 2005 was used in the experiments. The front engine with a transverse layout is equipped with four cylinders arranged in line with two valves per cylinder, four hydraulic tappets, and a neutral rotation frequency of motor about 900 rpm.

Considering a sequence of pistons from left to right of the vehicle, when the engine is started, pistons 1 and 4 move symmetrically, and pistons 2 and 3. Two explosions occur at each complete revolution that the shaft makes, one on pistons 1 or 4 and another on pistons 2 or 3. In this work, when failure in 1 piston is mentioned, it is referring to piston 1. When a failure in 2 pistons occurs, it means pistons 1 and 4 have failures. Misfire failure was induced by replacing a healthy piston with defective (inactive) spark plugs, which equates to 100% failure.

The acquisitions were made with the cell phone positioned in neutral over the car’s engine. Signals were acquired in 3 different scenarios, healthy condition, one-piston failure, and two-piston failure. In Figure 3, the vibration signals in the time domain are illustrated.

The acyclism frequency in the motor is twice the frequency of rotation of the motor, and both these frequencies are essential to understanding the Figure 4. It shows curves with the motor FFT signals for different fault conditions, in which the prominent peaks are highlighted. The signal’s fundamental frequency occurs at 30.18 Hz (acyclism frequency) for the motor to run normally. In this case, the motor is balanced, and the burst frequency becomes more evident in the vibration signal spectrum than the motor shaft rotation frequency, which is 15.09 Hz (905 rpm). When the pistons fail, the motor changes its rotational speed to 13.3 Hz (788 rpm), and its vibration increases, making the peak of this frequency more evident in the spectrum. In the exact figure, it is interesting to observe the frequency of 6.67 Hz in evidence because it is the explosion frequency of each piston individually (each piston explodes one time every two engine cycles, therefore half the rotation frequency).

There is a correlation between the FFT results of the article under review with the results obtained in [1] whose acquisitions were performed by another acquisition system in the same combustion engine of this work.

Through the vibration signals in the time domain using the Equation (Equation 8), the autocorrelation curves were generated for the different operating conditions of the motor using the detail D3 of the WMA, whose first cycle can be seen in Figure 5. It is possible to identify the autocorrelation coefficient (τ) with a healthy combustion engine, with one failed piston (1 FP), and with two failed pistons (2 FP), analyzing the decay time of the signals at half-height.

In Table 2, it is possible to identify the values of the correlation coefficient (τ) for each motor operating condition. From the respective values of τ the density of maxima values of the theoretical (Equation (Equation 9)) and the experimental values (Equation (Equation 10)) were obtained. The maximum relative error of 6.8% is evidence of the signal’s chaotic behavior.

Figure 6a–c illustrate the long-term autocorrelation function (Equation (Equation 7)) for vibration signals obtained with the healthy engine, with 1 piston failure and 2 piston failure respectively. In all scenarios, a long-term decay is noticed, which also shows the chaotic behavior of the signal [17].

Figure 7a–c illustrate the SAC-DM curve for the healthy combustion engine, with 1 faulty piston (1 FP) and 2 faulty pistons (2 FP). The difference between the graphs is in the window used to calculate the SAC-DM.

For the analysis via WMA, the D3 detail was used due to the composition of the signal frequencies. After several analyses, the statistical index’s standard deviation proved more coherent in the pattern change according to the operating conditions. Figure 8 illustrates the standard deviation of the vibration signal collection dataset for each WMA detail. In detail D3, the difference in the standard deviation value of the signals becomes more evident, proving the expected since its passband coincides with the explosion frequency of the pistons and engine rotation.

Similarly, Figure 9 illustrates the WMA detail D3 standard deviation curves over time for the three analyzed scenarios, using different acquisition windows.

To estimate both techniques’ accuracy, the maximum and minimum values of the healthy engine within the time acquired, with failure in 1 and 2 pistons, are observed. The accuracy was analyzed in 2 ways: diagnostic capability of the technique (can identify the healthy engine, one piston failure, and two pistons failure) and detection capacity (healthy motor or with some failure). The diagnosis is 100% accurate when there are no intercept points between the range and each curve’s maximum and minimum values. In the case of intersection points, the percentage concerning the total points on the curve is checked. When there is no intercept between the healthy motor or some fault, the detection is 100% accurate. Two lines are drawn in the graphs, the healthy borderline (HBL) and the diagnostic borderline (DBL) so that this can be better identified. For example: in Figure 7a, 21.85% of the points with 2-piston failures are below the line that indicates the maximum value of the 1-piston failure curve (diagnostic borderline is fixed at this value). In this case, we considered an accuracy of 78.15%. From the intersection between the range of values of the healthy motor or with some failure, it is possible to see that 9.25% of the values of the healthy motor curve are below the maximum failure value (healthy borderline is fixed at this value). Thus, it is considered that the accuracy of the measurement was 90.75%. The same logic applies to the other curves. Note that in Figure 7c and Figure 9c, which illustrate the curves of the SAC-DM and WMA values for windows that calculate 1.5 s and 5 s, respectively, there is no intersection between the maximum and minimum values of each curve, so it is considered 100% accurate for diagnosis and detection.

It is observed both in Figure 8 and Figure 9, a higher standard deviation for the vibration signal for the engine running with only one piston failure, which indicates more substantial signal energy in the predominant frequency of the D3 detail (25–12.5 Hz). This behavior can also be verified by looking at the amplitude of the peaks in this frequency window (Figure 4).

Table 3 compares the performance of the different techniques in terms of the diagnostic windows used based on curves in the Figure 7 and Figure 9,

From Table 3, a faster response can be seen when using the SAC-DM technique, enabling the detection of failure for 100% of the cases for a window of 1 s and the diagnosis for 100% of cases (severity of failure) for an acquisition window of 1.5 s. Despite a shorter response time, using the WMA approach, it was possible to obtain a diagnosis for 100% of the cases for a 5 s acquisition window.

It is possible to perceive that the effectiveness of both techniques is compatible with those of state the art (Table 1), being the SAC-DM the one that presents a lower necessary computational effort and with one of the most miniature acquisition windows for diagnostics, essential for critical failure applications.

## 6. Conclusions

The present work carried out a comparative approach between two time-domain techniques for misfire detection in combustion engines. For the first time, the SAC-DM technique was used to diagnose faults in combustion engines. An approach using the standard deviation of the D3 detail of a WMA was also implemented. Through the analyzes carried out, it was possible to identify that the diagnosis of misfire faults is feasible for 100% of the samples using a smartphone through the SAC-DM with 1-s sampling windows and for the WMA case for 5 s windows. It is worth noting that the SAC-DM is a technique with lower computational effort than the others focused on state-of-the-art based on the counting of peaks of a signal over time. The SAC-DM technique will be applied to diagnose other vehicle failures for future work.

## Figures and Tables

**Figure 1 sensors-22-05077-f001:**
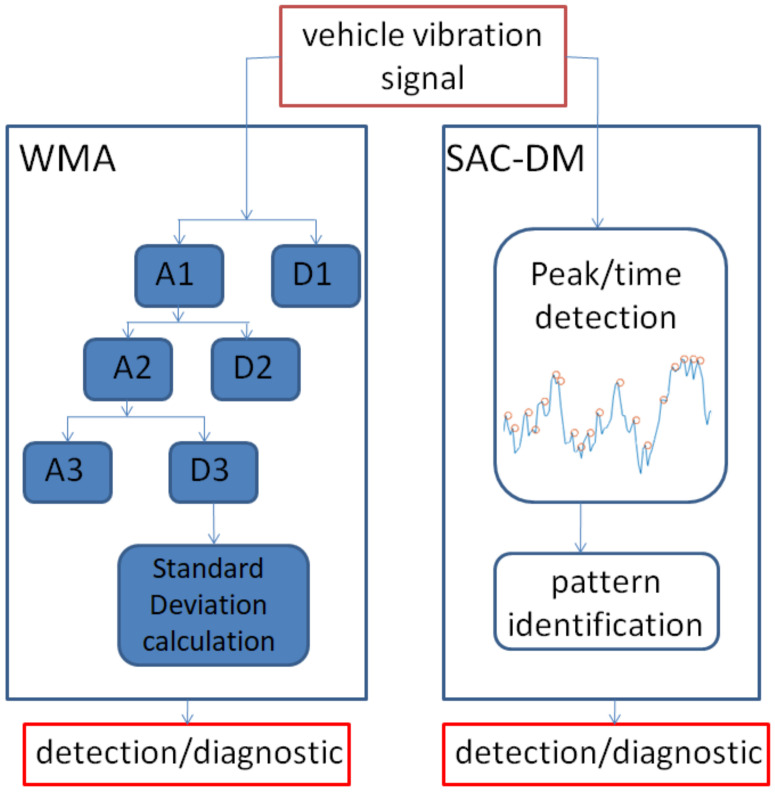
Block diagram of signal analysis methodology.

**Figure 2 sensors-22-05077-f002:**
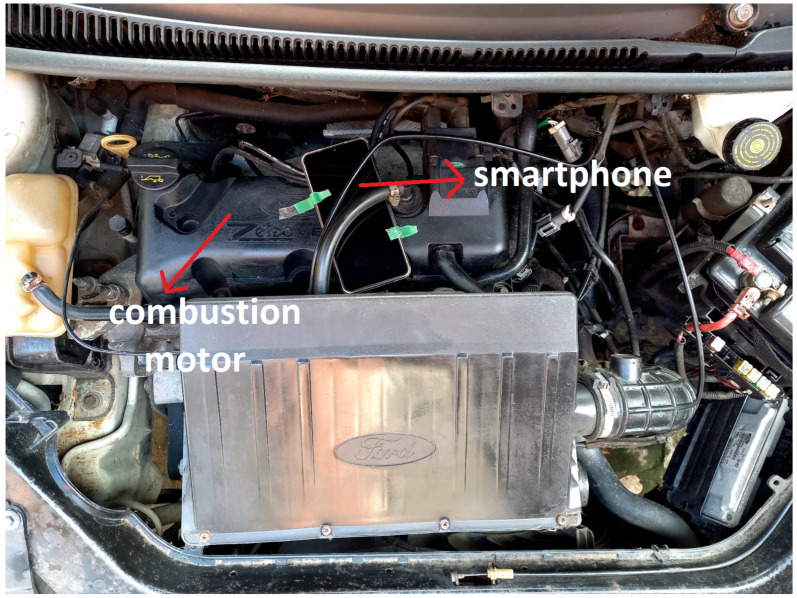
Smartphone positioning during acquisitions.

**Figure 3 sensors-22-05077-f003:**
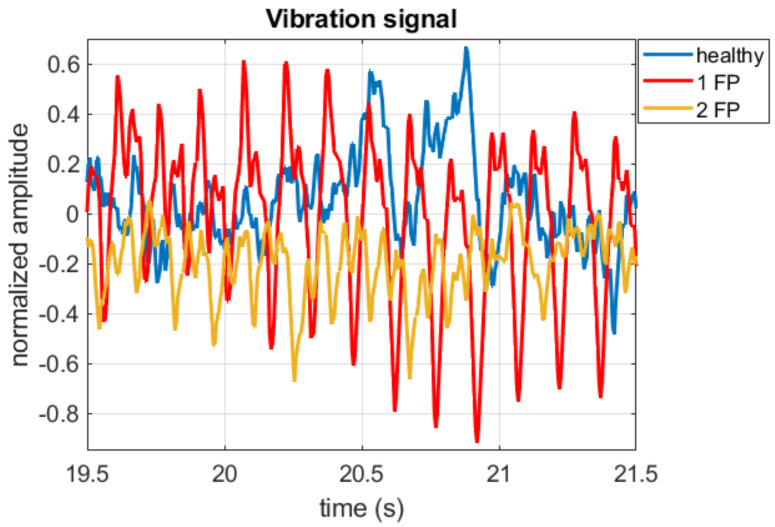
Vibration signals for healthy engine, 1 faulty piston (1FP), 2 faulty piston (2FP).

**Figure 4 sensors-22-05077-f004:**
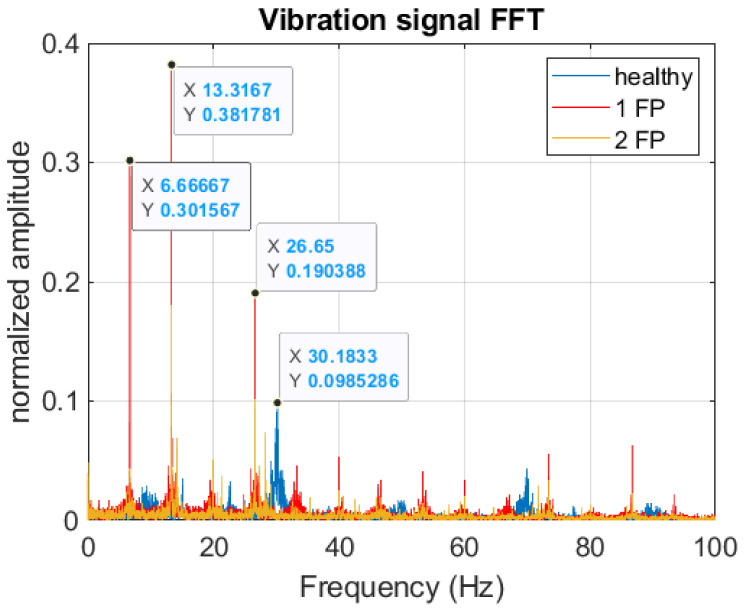
FFT signals for healthy engine, 1 faulty piston (1FP), 2 faulty piston (2FP).

**Figure 5 sensors-22-05077-f005:**
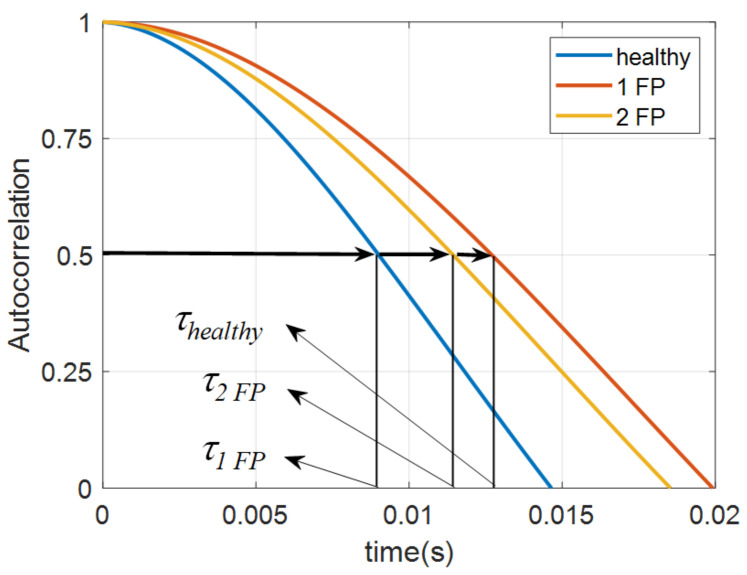
Autocorrelation for healthy engine, failing 1 piston (1 FP) and with failure of 2 pistons.

**Figure 6 sensors-22-05077-f006:**
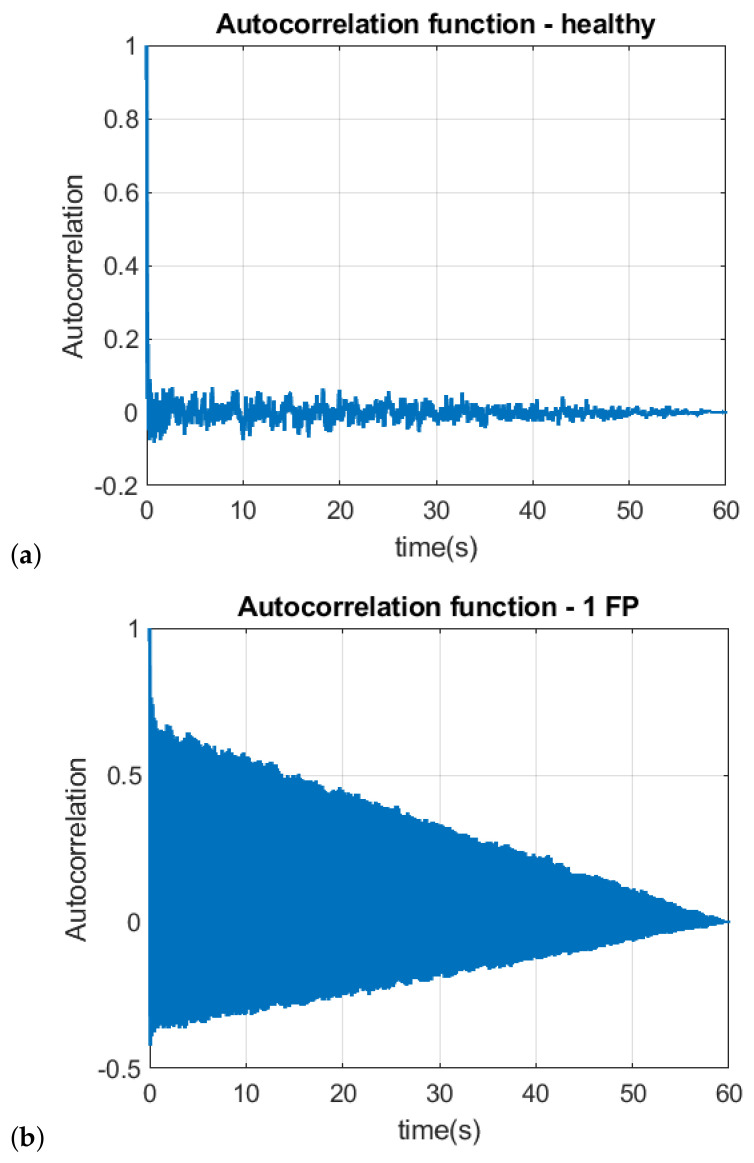
Autocorrelation for (**a**) healthy engine, (**b**) failing 1 piston (1 FP) and (**c**) with failure of 2 pistons.

**Figure 7 sensors-22-05077-f007:**
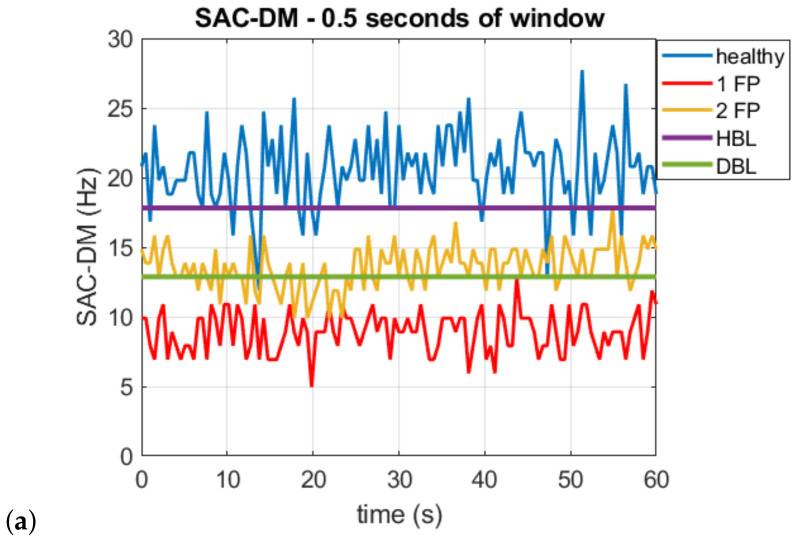
SAC-DM for a healthy engine with one faulty piston (1 FP) and two faulty pistons (2 FP). For time windows for SAC-DM calculation of (**a**) 0.5, (**b**) 1.0, and (**c**) 1.5 s.

**Figure 8 sensors-22-05077-f008:**
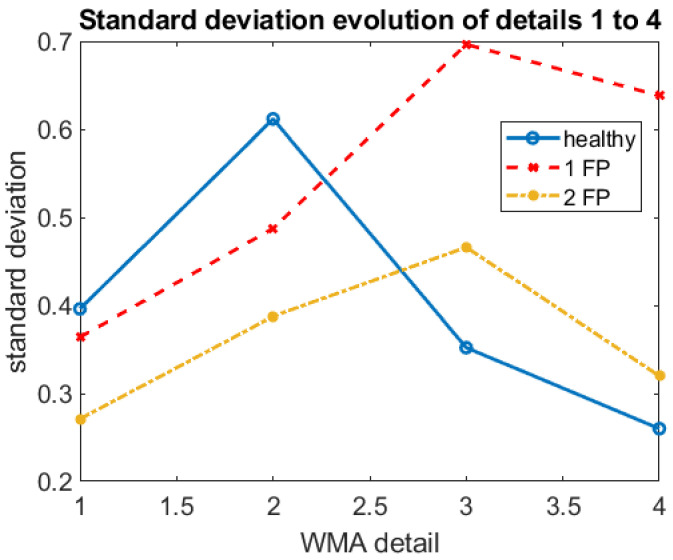
Evolution of standard deviation of samples as a function of WMA details for different combustion engine operating conditions.

**Figure 9 sensors-22-05077-f009:**
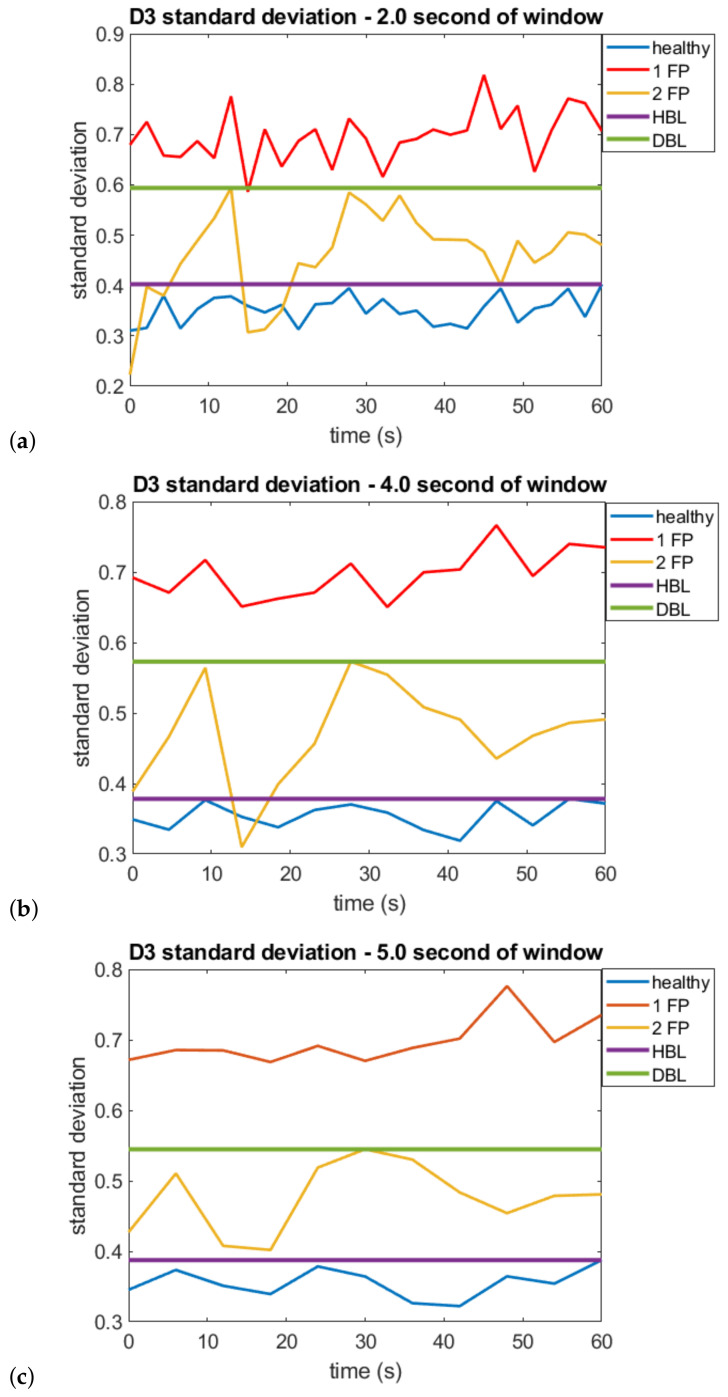
WMA Detail 3 standard deviation curves for the healthy engine, with one faulty piston (1 FP) and two faulty pistons (2 FP) for time windows for calculating the standard deviation of (**a**) 2, (**b**) 4, and (**c**) 5 s.

**Table 2 sensors-22-05077-t002:** Analysis of chaotic behavior of vibration signals.

Status	τ (msec)	ρ (Hz)	SAC-DM (Hz)	Relative Error (%)
Healthy	9.0	18.5	19.7	6.5
1 FP	12.6	13.2	13.3	0.8
2 FP	11.4	14.6	15.6	6.8

**Table 3 sensors-22-05077-t003:** Comparative chart of percentages for diagnosis and failure detection for the SAC-DM and WMA technique, for the different windows used.

	SAC-DM			Wavelet		
Time window (s)	0.5	1	1.5	2	4	5
Fault detection (%)	90.75	100.00	100.00	75.90	92.80	100.00
Fault diagnosis (%)	78.15	96.60	100.00	75.90	92.80	100.00

## Data Availability

Raw data are available from authors upon request.

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
