# Peer review of "Misfire Detection in Automotive Engines Using a Smartphone through Wavelet and Chaos Analysis"

_sensors, 2022, doi:10.3390/s22145077_

Round 1

Reviewer 1 Report

Wavelet Multiresolution Analysis (WMA) and Chaos Analysis using the density of maxima (SAC-DM) are used to identify misfare in a combustion engine of a working automotive vehicle. Experimental tests are also carried out for veriation. Results show that both methods havethe failure diagnosis for 100%.

Although the content is quite substantial, there are still some problems needs to be clarified.

1.The details of the two methods used need to be described more clearly. For example, to what extent does wma need to be decomposed? What is the peak count standard of sac-dm? What is the relationship between the two methods? Are they complementary? Why compare these two methods with ea other, rather than with other mainstream methods?

2. It is not clear that what is the main innovation in this paper, to improve the two methods or just only to apply these two methods to a new research object?

3. What is the special meaning of the marked data in Figure 7? Only one data is marked without explanation.

4. What is the RNA method in Table 1?

Author Response

All the comments are in the attached file.

Reviewer 2 Report

To improve the proposed paper, I suggest these modifications:

-             To ensure the reliability of the measured signal, it is necessary to correlate the signal obtained through the smartphone with another signal obtained with an acquisition system in which the accelerometer and the acquisition card are well-calibrated.

-             Table 1: The data processing techniques of the proposed article are mainly “Density of Maxima” and “Wavelet”. For the FFT technique, I suggest adding this reference:

Atef Hmida, Ahmed Hammami, Fakher Chaari, Mounir Ben Amar, Mohamed Haddar, Effects of misfire on the dynamic behavior of gasoline Engine Crankshafts, Engineering Failure Analysis 121 (2021) 105149

-             Table 1: How the accuracy (100%) of the proposed article is computed?

-             It is necessary to add the experimental setup (photos of the position of the smartphone with the engine).

-             How is the misfire defect introduced? What is the misfire percentage?

-             It is necessary to locate the number of the defective cylinders

-             Page 5, Line 105: I suggest replacing “explosion frequency” with “acyclism frequency”.

-             Figure 2: it is difficult to distinguish curves of 1 faulty piston (1FP), and 2 faulty pistons (2FP). I suggest changing the colors of the curves.

-             The title of figure 4 must be in English

-             Figures 5, 6, and 7 must be well interpreted. In figure 7, the authors should explain the reasons for crossing curves of healthy and 2FP (between 10s and 20s). In addition, the authors should explain the reason that the standard deviation of one faulty piston (1FP) is bigger than those of two faulty pistons (2FP).

-             References list: From 17 references, this is 5 self-citations (references 1, 2, 3, 11, and 16).

Author Response

(The authors gave the same response as above.)

Reviewer 3 Report

The paper is very interesting from a scientific point of view. The results are noteworthy and merit to be published. The paper is well structured and clear, I suggest improving English grammar, since there are a lot of mistakes. Besides, I think some recent relevant study works on engines should be added, to make the literature reviews more completed and help to the results analysis on the current state. For example, Fuel 287 (2021) 119418; Chemosphere 299 (2022) 134491; Applied Energy 255 (2019): 113800, etc..

Author Response

(The authors gave the same response as above.)

Round 2

Reviewer 1 Report

The manuscript is improved and can be accepted for publication in Sensors.

Reviewer 2 Report

The authors have ammended all the issues arised during the review, and I think that now the manuscript can be accepted for publication.